# GENERATIVE ENTITY NETWORKS: DISENTANGLING ENTITIES AND ATTRIBUTES IN VISUAL SCENES USING PARTIAL NATURAL LANGUAGE DESCRIPTIONS

## ABSTRACT

Generative image models have made significant progress in the last few years, and are now able to generate low-resolution images which sometimes look realistic. However the state-of-the-art models utilize fully entangled latent representations where small changes to a single neuron can effect every output pixel in relatively arbitrary ways, and different neurons have possibly arbitrary relationships with each other. This limits the ability of such models to generalize to new combinations or orientations of objects as well as their ability to connect with more structured representations such as natural language, without explicit strong supervision.

In this work explore the synergistic effect of using partial natural language scene descriptions to help disentangle the latent entities visible an image. We present a novel neural network architecture called Generative Entity Networks, which jointly generates both the natural language descriptions and the images from a set of latent entities. Our model is based on the variational autoencoder framework and makes use of visual attention to identify and characterise the visual attributes of each entity. Using the Shapeworld dataset, we show that our representation both enables a better generative model of images, leading to higher quality image samples, as well as creating more semantically useful representations that improve performance over purely dicriminative models on a simple natural language yes/no question answering task.

## 1 INTRODUCTION

The field of representation learning is motivated by the observation that the performance of machine learning algorithms depends heavily on the data representation used for learning. Good representations disentangle explanatory factors in the data, while others obfuscate such factors (Bengio et al., 2013). In the area of artificial intelligence, the hope of representation learning is that we can build models to *automatically* learn to identify and disentangle the underlying factors hidden in low-level perceptual data such as the pixels in images. This problem is fundamentally ill-posed, however, since the best representation is task dependent. None-the-less we can hope to find a representation which is generally useful for the kinds of tasks which humans want to perform. One such representation is the one used internally in the human thought process, which has proven to be useful for the wide variety of tasks that humans can perform successfully. While we cannot get at this representation directly, we can observe the way people describe concepts in natural language, and these natural language descriptions can provide us a window into it.

One area of representation learning which has seen significant progress in the last few years is generative modeling of images. With the advent of Variational AutoEncoders (VAEs), (Kingma & Welling, 2013; Rezende et al., 2014), and Generative Adversarial Networks (GANs), (Goodfellow et al., 2014), we are now able to generate low-resolution images that in some cases look very close to natural images. While existing work is partially motivated by more immediate applications in image processing and computer vision, much of the excitement stems from the belief that the ability to synthesize, or create the observed images implies a representation that has disentangled the important underlying concepts in the images signifying a deep understanding of the depicted scene (Radford et al., 2015).

While recent work has attempted to learn disentangled representations, e.g. Chen et al. (2016) and Bouchacourt et al. (2017), the representation in the best performing generative models of images are unstructured, with the dimensions of the latent vector fully entangled such that small changes to a single neuron can effect every output pixel in relatively arbitrary ways, and different neurons have possibly arbitrary relationships with each other.

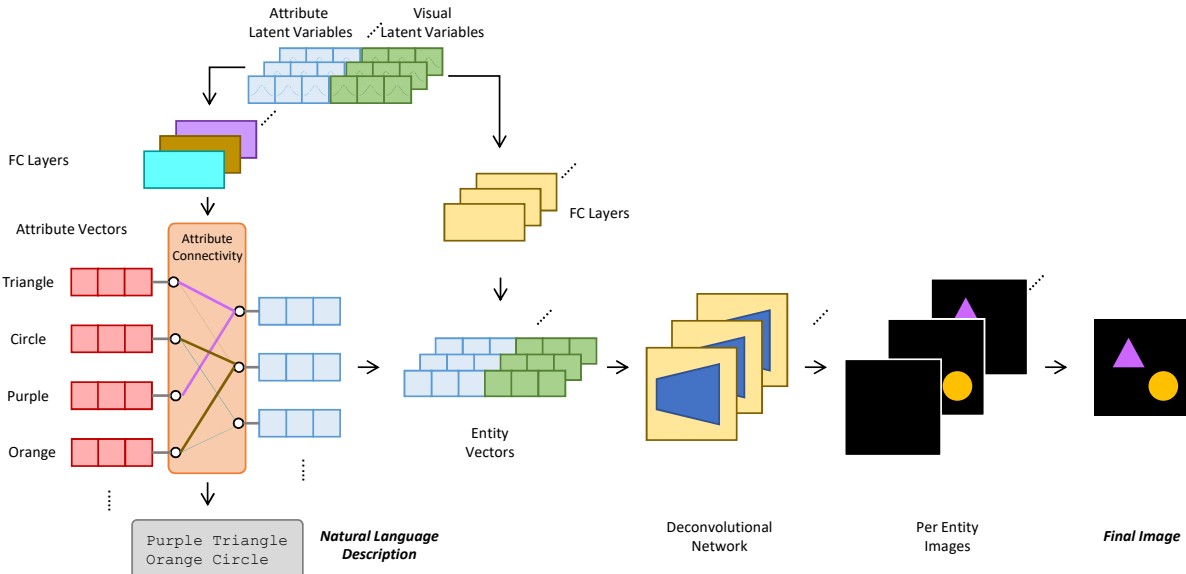

Figure 1: **Generative Decoder Model:** This figure shows the generative model used in Generative Entity Networks. The entity vectors are generated independently for each entity from two latent components. The *describable* component is generated as a weighted sum of the learned attribute vectors. The *indescribable* component is generated directly from a separate noise vector for each entity. The two components are concatenated together into the entity vector which is fed to a deconvoluational network to generate the part of the image associated with that entity. These partial images are then summed together to create the final output image.

In this paper we explore the use of natural language descriptions to help structure the latent representations for generative image models. Our hope is that the structure imposed by natural language is helpful for generalization and to anchor semantics in the latent representation. We believe this because individual words in a natural language description typically refer to some kind of entities, properties of an entity, or relationships between entities. Therefore representing entities and their attributes individually is a compact and accurate reflection of the world as humans reason and communicate about it.

When describing *visual scenes* the entities themselves are visual and typically are nouns such as *car* or *circle* which might have properties such as *color*, *model* or *size*. Such separation into entities often matches the generative structure of the image as well, with separate entities for each object in the scene. For the case of images, the pixels within a given entity will have a very tightly entangled relationship with each other, such as the different folds in the shirt of a person, while the relationship between pixels in different entities will be much less dependent, such as the pixels of two different people standing near each other. We would like to build on this intuition in order to create a generative model which generalizes better by disentangling entities whose resulting pixels are only weakly dependent.

We take a first step in this direction with a novel neural network architecture called *Generative Entity Networks* (GENs). GENs jointly generate natural language descriptions and images. GENs are based on the idea that both the natural language description and the observed pixels on the screen are generated by a set of latent entities or concepts, each represented by its own latent representation. Each entity, if observed, generates a noun phrase in the natural language, and/or a partial image in the visual modality. Each entity vector is generated from two components:

- **Describable component:** This represents the aspects of the entity which could be described in natural language such as their size or color. It is generated as a weighted combination of learned attributed vectors which are fixed to the words in our vocabulary.

- **Indescribable component:** This represents the visual details of the entity which are not described in natural language, such as as the intricate details of the shape, the lighting and the shadows. It is generated directly from a noise vector generated independently for each entity.

To learn the parameters of the generative model, we build on the VAE architecture, (Kingma & Welling, 2013; Rezende et al., 2014). We use a decoder network following the generative process as described above. For the

encoder we use a visual attention based encoder similar to Eslami et al. (2016a) and combine this with a natural language component.

While most recent work on generative image models has focused on the quality of the resulting images, our focus in this work is on the utility of the resulting latent representations. Thus we evaluate our model by using the learned latent representations to perform discriminative tasks from the ShapeWorld (**?**) dataset. This dataset contains noisy images of basic shapes with various attributes, such as color and shape, along with natural language descriptions of those images. We train our generative model on a part of the dataset that contains only partial natural language descriptions about the existence of various shapes in the scene. We then use the just the encoder of the VAE to generate the latent representations, and use these representations to perform standard discriminative tasks. Specifically, we test on both a discriminative version of the same task, as well a very different quantification task for which the generative model was never trained. We show that our generative representation significantly improves the performance above the baseline ShapeWorld models based on a state-of-the-art Recurrent Neural Network (RNN) and Convolutional Neural Network (CNN) architecture.

In summary, we make the following contributions:

- We propose a joint generative model of images and natural language descriptions with rich entity-based latent variable representation;
- We demonstrate improved performance at multiple image reasoning tasks on the ShapeWorld dataset.

## 2 BACKGROUND

*Variational autoencoders* (VAE) (Kingma & Welling, 2013; Rezende et al., 2014) are a class of deep latent variable models which specify a distribution over observable $\mathbf{x}$ as an infinite mixture,

$$p(\mathbf{x}) = \int p_\theta(\mathbf{x}|\mathbf{z}) \, p(\mathbf{z}) \, \mathrm{d}\mathbf{z}. \tag{1}$$

The form (1) allows for rich models because the conditional distribution $p(\mathbf{x}|\mathbf{z})$ is defined using deep neural networks having parameters $\theta$. The resulting marginal distribution $p(\mathbf{x})$ is an infinite mixture distribution which can capture higher order correlation structure efficiently.

Learning VAE models directly using maximum likelihood estimation with equation (1) is not possible due to the intractable integral in (1). However, Kingma & Welling (2013); Rezende et al. (2014) propose to combine *variational inference* approximations with amortization to provide a tractable and efficient bound on the marginal log-likelihood. This bound is the evidence lower bound (ELBO), defined as

$$\log p(\mathbf{x}) \quad \geq \quad \mathbb{E}_{\mathbf{z} \sim q_\omega(\mathbf{z}|\mathbf{x})} \left[ \log \frac{p_\theta(\mathbf{x}|\mathbf{z}) \, p(\mathbf{z})}{q_\omega(\mathbf{z}|\mathbf{x})} \right] \quad =: \quad \mathcal{L}(\mathbf{x}, \theta, \omega). \tag{2}$$

In (2) the auxiliary *inference network* $q_\omega(\mathbf{z}|\mathbf{x})$ approximates the true but intractable posterior $p(\mathbf{z}|\mathbf{x})$. To train the model we maximize $\mathcal{L}(\mathbf{x}, \theta, \omega)$ jointly over both $\theta$ and $\omega$ using a naive Monte Carlo approximation for the expectation in (2).

The VAE approach is very general and flexible and in our case the observations $\mathbf{x} = (\mathbf{I}, \mathbf{a})$ correspond to both an image $\mathbf{I}$ and a description of the scene $\mathbf{a}$ depicted in the image. We now describe our method as a way to impose rich prior knowledge on how entities relate to image content.

## 3 GENERATIVE ENTITY NETWORK

A generative entity network is a structured VAE in which the latent space is divided into a fixed number of latent entities. A multi-modal encoder takes an input image along with a description of one of the objects in the scene and encodes to a distribution over latent codes for each object. The latent codes are decoded with a multi-modal decoder that outputs the parameters of a pixel distribution, as well as a set of distributions over object descriptions. We make the distinction between *attribute* latent variables, which encode features of the objects that are describable by language, and *visual* latent variables, which encode the remaining visual information required to represent the object in an image. In ShapeWorld the attributes of objects that are language-describable are objects' shapes and colors, whereas objects' positions, scale and orientation are not described in captions.

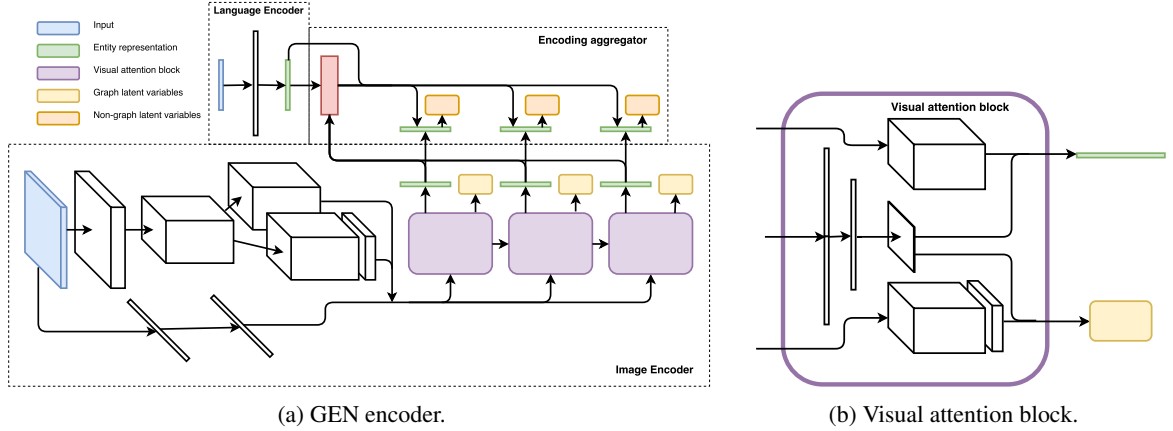

(a) GEN encoder.

(b) Visual attention block.

Figure 2: **Encoder Model**

### 3.1 IMAGE-LANGUAGE ENCODER

The GEN encoder (Figure 2a) consists of separate image and language processing modules, as well as an aggregator which reconciles objects distributions from both modalities.

**Image encoder.** In order to encode relevant parts of the image in separate latent entity variables, we employ a recurrent visual attention mechanism. Initially the image is processed to obtain two sets of feature maps $\mathbf{F}^A$, $\mathbf{F}^V$, where the first is used to encode attribute latent variables, and the second is used to encode visual latent variables. These feature maps are fed into a series of recurrent attention blocks, where at every step an LSTM outputs a soft-attention mask $\mathbf{M}_k$ which is applied to both sets of feature maps to output localized object representations:

$$\mathbf{a}_k^I = \sum_{ij}[\mathbf{M}_k]_{ij}[\mathbf{F}^A]_{ij} \tag{3}$$

$$\mathbf{v}_k = \sum_{ij}[\mathbf{M}_k]_{ij}[\mathbf{F}^V]_{ij} \tag{4}$$

Finally each visual object representation $\mathbf{v}_k$ is passed through an MLP to obtain the parameters of the approximate posterior distribution over that object's visual latent variables.

**Language encoder.** A fully-connected MLP takes the language input and outputs an attribute vector $\mathbf{a}^L$ of the same dimensionality as the visual attribute vectors $\mathbf{a}^I$. Note that as we use a parsed-binary language input there is no need for sequential processing using e.g. an RNN, however it would be straight-forward to incorporate a language encoder of this kind in order to handle natural language inputs.

**Multi-modal aggregator.** We assume that the encoder takes as input a description of a single object as well as an image which may have multiple objects. This poses a challenge, as we don't know which of the objects encoded by the image encoder corresponds to the object described in the input caption. We compute the agreement $g_k$ between descriptions from the two modalities an agreement operator that computes the dot product between the object attributes encoded by the language encoder, and each of the object attributes encoded by the image encoder: $g_k = \mathbf{a}_k^{I^T}\mathbf{a}^L / \sum_k \mathbf{a}_k^{I^T}\mathbf{a}^L$. We obtain an aggregated representation for each objects attributes by taking an agreement-weighted average of language and image attribute vectors $\mathbf{a}^L$ and $\mathbf{a}^I$:

$$\hat{\mathbf{a}} = (\mathbf{a}^I + g_k\mathbf{a}^L)/(1 + g_k) \tag{5}$$

### 3.2 MULTI-MODAL DECODER

The basic structure of the GEM decoder (Figure 2a) is that latent objects are decoded independently using the same network. Visual latent variables only decode to image outputs, whereas attribute latent variables influence both the visual and language outputs. The attribute latent variables for a particular object decode to a vector

of attribute assignments, which represent which attributes are present in that object. After the latent objects are decoded to a collection of object images, they are composited to form the decoded output image.

**Object attributes.** We augment the decoder network with a collection of $A$ object attributes $\mathbf{h}_a$, each of which is a vector representation of a particular attribute, like shape or color.

**Attribute assignments.** Attribute latent variables $\mathbf{z}_k^A$ for the $k$'th entity decode directly to a vector of attribute assignments $\mathbf{A}_k$ via an MLP with sigmoid outputs: $\mathbf{A}_k = \sigma(\mathrm{MLP}(\mathbf{z}_k))$. The attribute assignment vector is used in two ways: First, it directly represents the parameters of the decoded language vector. Secondly, for a particular object, an object attribute representation $\mathbf{a}_k$ is obtained by taking a weighted sum over the attributes, using the attribute assignments as weights:

$$\mathbf{a}_k = \sum_{a=1}^{A} [\mathbf{A}_k]_a \odot \mathbf{h}_a. \tag{6}$$

**Visual latent variables.** The visual latent variables $\mathbf{z}_k^V$ for entity $k$ are decoded to an object visual representation using an MLP: $\mathbf{v}_k = \mathrm{MLP}(\mathbf{z}_k^V)$. The visual object representation $\mathbf{v}_k$, and object attribute representation $\mathbf{a}_k$ are concatenated to form a combined object representation $\mathbf{c}_k = [\mathbf{v}_k, \mathbf{a}_k]$.

**Drawing entities.** For each entity the GEN decoder renders an image $\hat{\mathbf{x}}_k$ by mapping the combined entity representation $\mathbf{c}_k$ through a series of reshaping, convolutional and upsampling layers. An aggregated image is obtained by summing the entity images, along with a black background $\hat{\mathbf{x}} = \mathbf{b} + \sum_k \mathbf{x}_k$.

### 3.3 Learning

We train the parameters of the encoder and decoder simultaneously by maximizing a variational lower bound on the model log-likelihood. We use a Bernoulli pixel distribution, where each pixel is modeled as independent given the latent variables. For the binary language we also use an independent Bernoulli distribution. However as the model outputs language predictions for multiple objects, and yet only one object is described in language per scene, we maximize over assignments of predicted language to the true caption:

$$p(\mathbf{I}|\mathbf{z}) = \mathbb{B}(\mathbf{I}|\hat{\mathbf{I}}(\mathbf{z})) \tag{7}$$

$$p(\mathbf{L}|\mathbf{z}) = \max_k \mathbb{B}(\mathbf{L}|\hat{\mathbf{L}}_k(\mathbf{z})) \tag{8}$$

### 3.4 Auxiliary visual question answering network

Given a trained GEN, it is possible to use the encoder to infer properties of objects present in a scene. This facilitates effective learning of auxiliary tasks such as visual question answering, with a representation that is already factored into distinct objects. The ShapeWorld agreement tasks consist of input pairs of images and captions, and the task is to predict whether or not the caption is a true description of the image.

**GEN image Encoder.** Input images are processed using the pre-trained GEN image encoder, outputting representations for each object $\mathbf{o}_k = \mathbb{E}[q(\mathbf{z})]$.

**Caption embedding.** Captions $\mathbf{c}$ are processed using an LSTM outputting a caption embedding $\mathbf{e} = \mathrm{LSTM}(\mathbf{c})$.

**Reasoning.** The caption embedding is concatenated to every object embedding and processed independently using a shared MLP: $\mathbf{r}_k = \mathrm{MLP}([\mathbf{o}_k, \mathbf{c}])$ to obtain caption-conditioned object representations. We aggregate across objects using a symmetric element-wise function. In practice we found the element-wise max to be effective choice: $\mathbf{a} = \max_k \{\mathbf{r}_k\}_k$. A final MLP produces the logits of the agreement estimate as a function of the aggregated object-caption pairs $l = \mathrm{MLP}(\mathbf{a})$. We train the reasoning module to minimize the binary cross-entropy loss for the the predicted and true agreements.

## 4 Experimental Settings

### 4.1 Datasets

**ShapeWorld** We perform our primary evaluation on the ShapeWorld dataset (Kuhnle & Copestake, 2017). This is an automatically generated dataset of images containing primitive shapes with a natural language description

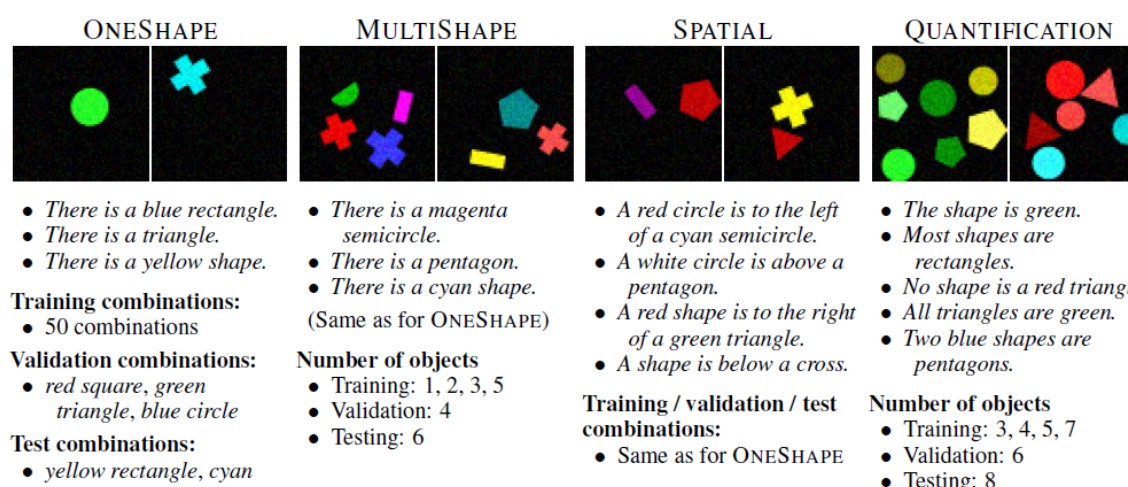

Figure 3: **ShapeWorld Dataset:** Examples from the ShapeWorld dataset. This figure is recreated from Kuhnle & Copestake (2017).

for each image partially describing the set of shapes visible in the image. The shapes have 3 primary attributes: color, geometric shape, and location. The natural language describes only the primary attributes. The shapes also have 5 secondary attributes: orientation, size, shade, distortion, and noise level. These attributes are never discussed in the natural language. The distortion level controls the width vs. height property for rectangles and ellipses, and the shading will change the color within some range from the primary color attribute. The dataset contains 8 shapes, and seven colors, and all other attributes are continuous. An image is generated by first choosing the number of shapes, and then independently randomly choosing each attribute for each shape. A single natural language description is generated for each image based on the four different tasks outlined in Figure 3. Note that the location attribute is only referenced in the Spatial task, while the color and shape attributes are referenced in all four tasks. Each natural language description is marked as either *True* or *False*, with an equal number of each type.

**SimpleShapes** In order to show some of our qualitative evaluations more cleanly, we also performed experiments on a simple dataset we generated, called SimpleShapes. The details of this dataset are discussed in Appendix A.

### 4.2 BASELINES

We compare against the four baselines from Kuhnle & Copestake (2017). These all use either an LSTM or a CNN to process the text, and a CNN to process the image.

**LSTM-only** This baseline only looks at the text itself, and so the only reason this model can do better than random is because of biases in the dataset itself. From the results, we can see that this dataset is much less biased than other Visual Question Answering Datasets (**?**)jabri2016revisiting).

**CNN+LSTM:Mult** This baseline obtains an embedding of the text from the LSTM and performs a pointwise multiplication with the CNN representation of the image.

**CNN+CNN:HCA-par, CNN+CNN:HCA-alt** The proposed methods of Kuhnle & Copestake (2017), using a CNN for the text and a separate CNN for the image, along with one of two different novel co-attention mechanisms (*parallel* or *alternating*). In Kuhnle & Copestake (2017) both methods performed comparable and significantly outperforming the other methods for some tasks.

### 4.3 GEN TRAINING SETUP

We train our GEN model only on data from the MultiShape task. As our model does not currently have an simple way to incorporate negative information, we train the model on only the samples where the natural language description is marked as *True*. To feed the natural language data into our model, we first transform each sentence

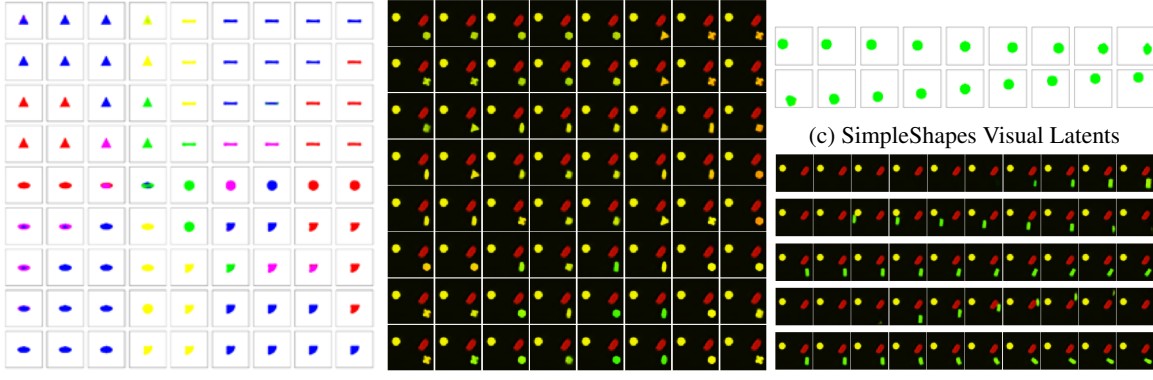

(c) SimpleShapes Visual Latents

(d) ShapeWorld Visual Latents

(a) SimpleShape Attribute Latents     (b) ShapeWorld Attribute Latents

Figure 4: **Latent Variable Interpolations:** shows how the image changes as we smoothly change the latent variables for one of the entities. Figures (a) and (b) show changes in the attribute latent variables. We can easily see in (a) that each shape and each color occupies a continuous region of the latent space, with relatively sharp changes between these regions. Note that we should not expect the division between the color and shape semantic attributes to align to the two latent dimensions since the GEN model leaves the encoding of the attribute dimensions completely entangled for a given entity. Figures (c) and (d) show changes in the visual latent variables. We can see easily in (c) that the shape and color remains constant while only the position changes. We can see from Figures (b) and (d) that in the ShapeWorld object existence is controlled by both the attribute latent variables and the visual latent variables.

into a binary bag of words representation after dropping the non-content words, i.e. words that doesn't describe one of the attributes. All of our qualitative results are generated from the resulting model.

The quantitative results are generated by combining the trained encoder from the GEN with a separate auxiliary model for each task as described in Section 3.4. The auxiliary models are trained discriminatively to predict *True* or *False*, given the image and the natural language sentence. When training the auxiliary model, the parameters of the GEN encoder are fixed, and not updated. Thus, the resulting encoder representation of the image and the natural language is fixed across all four tasks.

## 5 EVALUATION

### 5.1 QUALITATIVE EVALUATION

In order to confirm that our model learns as expected we also performed a series of qualitative experiments. Two of these are shown here, while in Appendix B we compare samples of our model to samples of a standard fully entangled VAE model, as well as showing the quality of the image reconstructions.

**Latent Variable Interpolations** Figure 4 shows the effect of smoothly changing the various latent variables for a both a GEN model trained on the SimpleShapes dataset as well as one trained in the ShapeWorld dataset. The SimpleShapes model was trained with only two latent dimensions for the attribute latent vector, to allow us to easily see the entire latent space. The ShapeWorld model had many more latent dimensions so we can only observe a small fraction of the latent space.

**Attention Maps** Figure 5 shows the attention maps generated by the encoder. We can see clearly that the encoder is appropriately attending to individual shapes, and aligning them with the appropriate natural language as we had hoped.

### 5.2 QUANTITATIVE EVALUATION

The main goal of our work is generated disentangled latent representations that can improve performance on a variety of downstream tasks. To this end, we train our GEN model on the MultiShape task, and use the learned

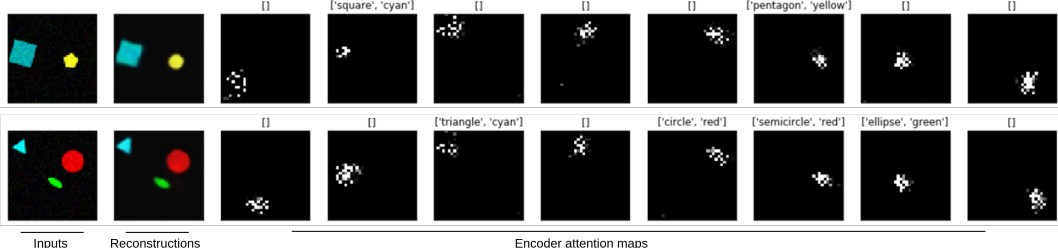

Figure 5: **Visual Attention Maps:** This figure shows both the visual attention map and the thresholded language prediction for each step in the recurrent encoder process. We can see that the attention maps are focused in the right area and aligned to the appropriate language descriptions. Occasionally a single shape in the image ends up a two separate entities in the model such as the red circle and red semi-circle in the second example.

| Method | OneShape | MultiShape | Spatial | Quantification |
|---|---|---|---|---|
| LSTM-only | 51 / 46 / 50 | 62 / 67 / 67 | 52 / 51 / 50 | 57 / 57 / 56 |
| CNN+LSTM:Mult | 81 / 70 / 66 | 72 / 71 / 72 | **72 / 71 / 69** | 71 / 68 / 68 |
| CNN+CNN:HCA-par | 90 / 77 / 78 | 72 / 71 / 69 | 63 / 65 / 64 | 76 / 77 / 78 |
| CNN+CNN:HCA-alt | 92 / 81 / 77 | 71 / 68 / 68 | 54 / 52 55 | 74 / 77 / 78 |
| GEN | **98 / 85 / 82** | **96 / 94 / 90** | 63 / 59 / 60 | **91 / 85 / 83** |

Table 1: **ShapeWorld Accuracy Results:** Accuracy on the train/validation/test sets for the four ShapeWorld tasks. Baseline results are reported as in Kuhnle & Copestake (2017). We see that the GEN improves performance for all tasks except spatial relations. Note that encoder component of the GEN model was trained only on data from the MultiShape task, yet the same representations result in improved performance on the OneShape and the Quantification task.

latent representations as the input to separate auxiliary models which are trained for each of the four ShapeWorld tasks.

We can see from Table 1 that the GEN model outperforms all the baselines on the OneShape, MultiShape and Quantification tasks. This shows that the learned latent representations generalize quite well across tasks since they enable superior performance on both the OneShape and the Quantification tasks. On the Spatial task the GEN model is outperformed by the CNN+LSTM model. We believe that this is because the data from the MultiShape task on which the GEN representations were trained, does not contain any information in the natural language about location, and location is critical to the Spatial task. The location information must still be encoded in the latent representation in order to facilitate correct image reconstruction, but it is entangled with all of the other secondary attributes, making it more difficult for the auxiliary model to utilize it.

# 6 RELATED WORK

The idea of using more structured latent representations in generative modeling has been broadly explored before. We now provide a brief overview of related work and emphasize how our method improves on these works.

## 6.1 STRUCTURED LATENT REPRESENTATIONS IN VAES

The *structured VAE* approach of Johnson et al. (2016) uses models composed of two parts: a graphical model for latent variables and a neural network model for an observation likelihood. They propose hybrid methods for efficient inference in case the graphical model structure allows for structured mean field approximations based on conjugacy. As such their approach combines key advantages of previous graphical model inference approaches with the expressive power of neural networks. Our model does not use a graphical model to specify the latent space and instead achieves expressivity through rich deterministic but entity-aware transformations of the latent representations. Combining the structured VAE approach with our expressive model is a useful extension of our work.

*Variational graph encoders* Kipf & Welling (2016) are extensions of VAE models for modeling distributions over undirected graphs. Like in our approach each entity/node is separately represented by the model, but in our case, entities relate to an additional image and our model needs to relate entities to image content. Similarly, the DRAW network of Gregor et al. (2015) implicitly models multiple entities such as digits using recurrence and attention models. However, their latent representation does not directly make the individual entities accessible and they do not generate natural language descriptions. Eslami et al. (2016b) does make entities individually accessing in a generative model of images based on VAEs and recurrent neural networks, however, their representation of each entity is not language-based and therefore largely opaque.

Our model uses entities as semantically meaningful latent representation; recent work instead attempts to *learn* disentangled representations in VAEs using weak supervision Bouchacourt et al. (2017). To enable reasoning over richer entities, we imagine a suitable extension of our work is to use this recent work to learn a high-dimensional free-form continuous representation describing the entity.

## 6.2 VISION AND LANGUAGE

There has been a significant amount of work done in connecting natural language and vision. We break this into three categories based on the type of the models used.

**Discriminative Models** The largest area of work on language and vision is in discriminative models. The most similar work to ours in the discriminative space is the Visual Genome(Krishna et al., 2017) project. The motivation behind this project is very similar to ours, predicting a latent entity-relation graph for an image. However, they attempt to manually specify these graphs through a large-scale annotation effort. These kinds of annotations are very expensive to obtain, making it difficult to scale such an effort broadly. Thus our motivation is to eventually be able to learn a similar latent graph representation from only partial natural language descriptions instead, which are much easier and natural to obtain. Beyond this, the most well known work in vision and language is on the COCO dataset (Lin et al., 2014) and the Visual Question Answering dataset (Antol et al., 2015). In a similar vein to the Visual Genome, Teney et al. (2016) also use graph-structured representations for visual question answering, however they work on the subset of the dataset which consists of abstract scenes, where such graphs are readily available, avoiding the need for annotation. The state-of-the-art models on the full datasets are discriminative neural network models combining CNNs and RNNs similar to the baseline models we outperform on the ShapeWorld dataset.

**Reinforcement Learning Models** There has also been some work in connecting language to vision in the reinforcement learning setting. Both Oh et al. (2017) and Hermann et al. (2017) present models where an agent is rewarded for the successful execution of natural language instructions. While this work is related, the reinforcement learning setting is quite different from ours.

**Generative Models** The most closely related work to ours involves generative models of vision and language. We build directly on the work of Eslami et al. (2016b), using a model very similar to theirs for our encoder. Similarly, the model of Greff et al. (2016) can learn to separate entities in an unsupervised manner, and the $\beta$-VAE model of Higgins et al. (2016) tries to learn a disentangled latent representation in a purely unsupervised manner. However, none of this work uses any natural language information, and so their learned disentanglement of the latent space does not necessarily capture the same semantic attributes important to humans. In (Higgins et al., 2017) they extend the $\beta$-VAE work to include a small amount of compositional supervision. Similarly, Vedantam et al. (2017) also attempted to learn a compositional model of image attributes. However, neither of these models has a concept of separate entities which each have their own attributes and locations, and so can only handle images that contain a single conceptual entity (which may have many parts). Reed et al. take a more heavily supervised direction. They learn to generate images from textual descriptions, but rely heavily on the strong supervision of object locations in order to obtain good performance. In contrast, our work relies only on partial natural language descriptions.

## 7 CONCLUSION AND FUTURE WORK

We proposed a novel probabilistic model for learning from both language and images. Our model directly models multiple entities through per-entity latent variables. We achieve this by using a joint generative model of image and language information which uses an effective attention mechanism to reason about one entity at a time.

Beyond providing a more natural representation, our model is also more accurate in tasks: we demonstrated large gains in task performance on the ShapeWorld data set over strong baseline methods.

In the future we will apply our model to larger visual and language domains towards the goal of modeling entire natural-language conversations about the world. To this end, we plan to extend our model in to ways: 1. adding a model of dependencies between multiple entities; and 2. adding more than one view on the world, i.e. modeling separate agents perceiving different parts of the same world.

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

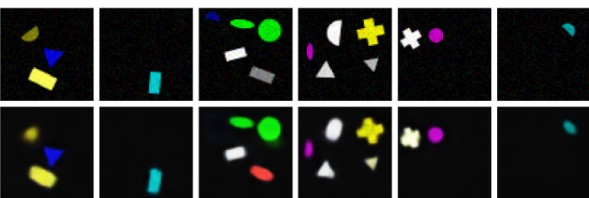

Figure 6: **Image Reconstructions:** This figure shows samples from the ShapeWorld dataset along with the reconstructions from our model. As with all VAE based models, the reconstructions are slightly blurry but otherwise relatively faithful. When the model does make mistakes these tend to be discrete in nature, such as the dropped half-circle, and the incorrectly colored rectange in the third image.

## APPENDIX A    SIMPLESHAPES DATASET

The SimpleShapes dataset consists of a single shape in each image with shapes varying only in the primary attributes of geometric shape, color and location. The secondary attributes considered in the ShapeWorld were all held fix, and the images did not contain any noise. The natural language descriptions always describe both the color and geometric shape of the observed shape but do not include the location.

## APPENDIX B    ADDITIONAL QUALITATIVE RESULTS

Here we show both samples from our model as well as image reconstructions.

**Reconstructions**    Figure 6 shows the quality of the image reconstrutions. We can see that the GEN is able to faithfully reconstruct most of the relevant attributes of the shapes, including their color, shape, size, location, and orientation.

**Image Samples**    Figure 7 compares unconditioned samples of our model to those from a standard fully entangled VAE on the SimpleShapes dataset. We can see that the traditional VAE struggles to generate identifiable shapes even on this relatively simple dataset. Figure 9 shows unconditioned samples from the ShapeWorld dataset. We can see that while these samples are quite blurry, the model is still generating distinct shapes, each with a distinct color. The most distinct shapes, such as the cross and the triangle are distinguishable, while the more similar shapes such as the the ellipse and the rectangle are harder to distinguish. As the standard VAE model struggled even on the SimpleShapes data set, we do not show standard VAE results for the ShapeWorld dataset.

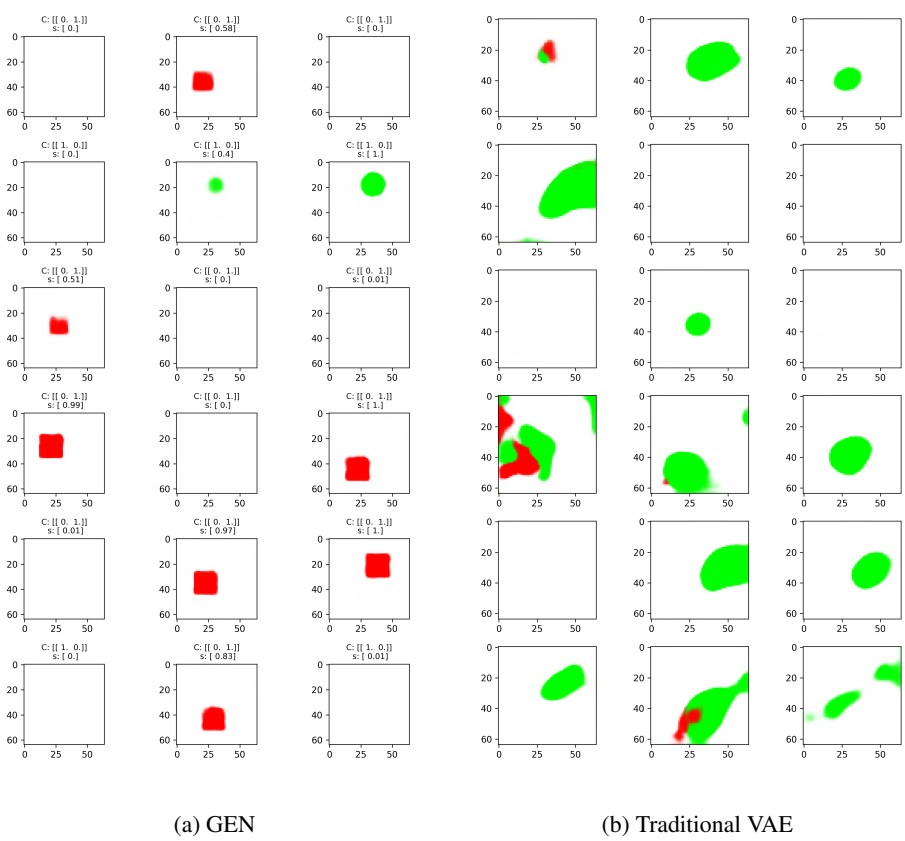

(a) GEN       (b) Traditional VAE

Figure 7: **Unconditioned SimpleShape Samples:** This figure compares the samples from the GEN model, and a traditional fully entangled VAE model. We can see that the GEN model is able to recognize and draw the shapes correctly, and consistently color each shape, while the traditional entangled model cannot consistently draw any of the shapes, and mixes together the colors as well.

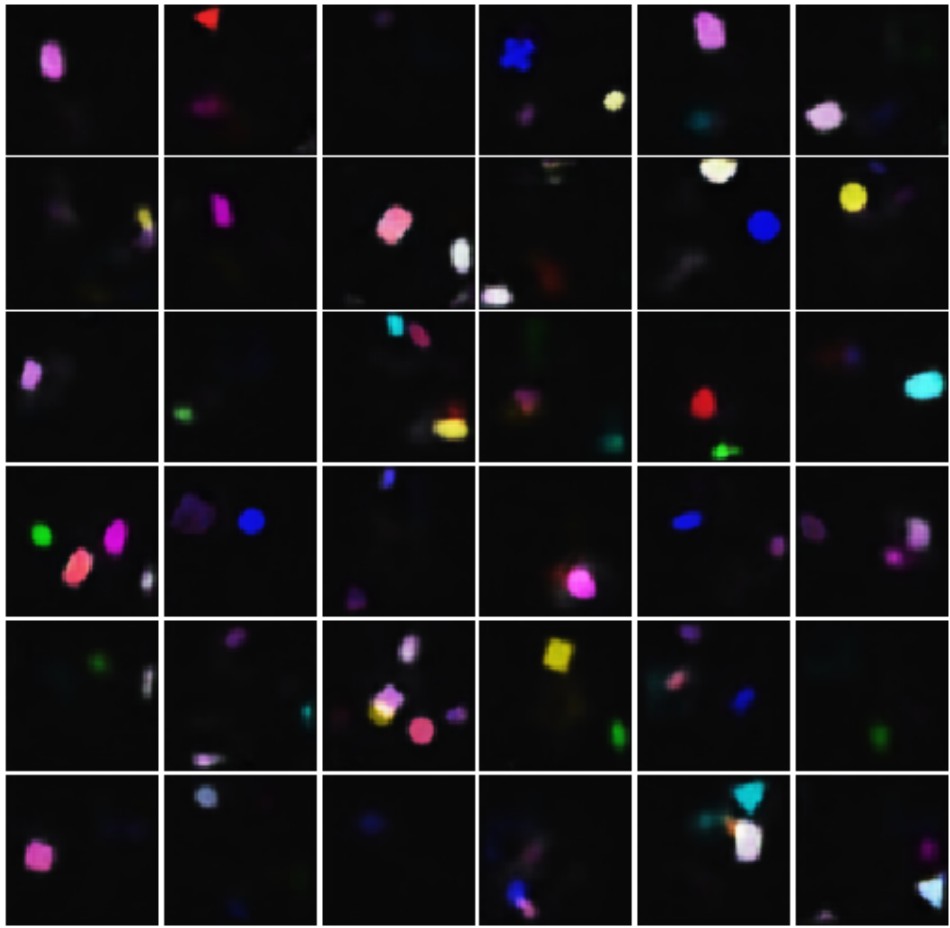

Figure 8: GEN ShapeWorld Samples

Figure 9: **Unconditioned ShapeWorld Samples**

