# OpenReview forum: "Generative Entity Networks: Disentangling Entitites and Attributes in Visual Scenes using Partial Natural Language Descriptions"
_ICLR.cc/2018/Conference — Reject_

### Official Review · AnonReviewer2 · 2017-11-27
**The paper needs to be polished, and more experimental evaluations are required.**

**Rating:** 4
**Confidence:** 4

**Review:**

This paper presented a Generative entity networks (GEN). It is a multi-view extension of variational autoencoder (VAE) for disentangled representation. It uses the image and its attributes.  The paper is very well motivated and tackles an important problem. However, the presentation of the method is not clear, the experiment is not sufficient, and the paper is not polished.

Pros:
1. This paper tackles an important research question.
Learning a meaningful representation is needed in general. For the application of images, using text description to refine the representation is a natural and important research question.

2. The proposed idea is very well motivated, and the proposed model seems correct.

Cons and questions:
1. The presentation of the model is not clear.
Figure 2 which is the graphic representation of the model is hard to read. There is no meaningful caption for this important figure.  Which notation in the figure corresponds to which variable is not clear at all.  This also leads to unclarity of the text presentation of the model, for example, section 3.2. Which latent variable is used to decode which part?

2. Missing important related works.
There are a couple of highly related work with multi-view VAE tracking similar problem have been proposed in the past. The paper did not discuss these related work and did not compare the performances. Examples of these related work include [1] and [2] (at the end of the review).
Additionally, the idea of factorized representation idea (describable component and indescribable component) has a long history.  It can be traced back to [3], used in PGM setting in [4] and used in VAE setting in [1]. This group of related work should also be discussed.

3. Experiment evaluation is not sufficient.
Firstly, only one toy dataset is used for experimental evaluations. More evaluations are needed to verify the method, especially with natural images.
Secondly, there are no other state-of-the-art baselines are used. The baselines are various simiplied versions of the proposed model. More state-of-the-art baselines are needed, e.g. [1] and [2].

4. Maybe overclaiming.
In the paper, only attributes of objects are used which is not semi-natural languages.

5. The paper, in general, needs to be polished.
There are missing links and references in the paper and un-explained notations, and non-informative captions.

6. Possibility to apply to natural images.
This method does not model spatial information. How can the method make sure that  simple adding generated images with each component will lead to a meaningful image in the end? Especially with natural images,  the spacial location and the scale should be critical.

[1] Wang, Weiran, Honglak Lee, and Karen Livescu. "Deep variational canonical correlation analysis." arXiv preprint arXiv:1610.03454 (2016).
[2] Suzuki, Masahiro, Kotaro Nakayama, and Yutaka Matsuo. "Joint Multimodal Learning with Deep Generative Models." arXiv preprint arXiv:1611.01891 (2016).
[3] Tucker, Ledyard R. "An inter-battery method of factor analysis." Psychometrika 23.2 (1958): 111-136.
[4] Zhang, Cheng, Hedvig Kjellström, and Carl Henrik Ek. "Inter-battery topic representation learning." European Conference on Computer Vision. Springer International Publishing, 2016.

---

### Official Review · AnonReviewer3 · 2017-11-27
**Extension of Attend Infer Repeat to include "language", evaluated on a question answering task. Approach is evaluated on a image caption agreement task.**

**Rating:** 5
**Confidence:** 4

**Review:**

**Summary**
The paper proposes an extension of the attend, infer, repeat generative model of Eslami, 2016 and extends it to handle ``visual attribute descriptions. This straightforward extension is claimed to improve image quality and shown to improve performance on a previously introduced image caption ranking task. In general, the paper shows improvements on an image caption agreement task introduced in Kuhnle and Copestake, 2017.  The paper seems to have weaknesses pertaining to the approach taken, clarity of presentation and comparison to baselines which mean that the paper does not seem to meet the acceptance threshold for ICLR. See more detailed points below in Weaknesses.

**Strengths**
I like the high-level motivation of the work, that one needs to understand and establish that language or semantics can help learn better representations for images. I buy the premise and think the work addresses an important issue.

**Weakness**

Approach:
* A major limitation of the model seems to be that one needs access to both images and attribute vectors at inference time to compute representations which is a highly restrictive assumption (since inference networks are discriminative). The paper should explain how/if one can compute representations given just the image, for instance, say by not using amortized inference. The paper does propose to use an image-only encoder but that is intended in general as a modeling choice to explain statistics which are not captured by the attributes (in this case location and orientation as explained in the Introduction of the paper).

Clarity:
* Eqn. 5, LHS can be written more clearly as \hat{a}_k.

* It would also be good to cite the following related work, which closely ties into the model of Eslami 2016, and is prior work:

Efficient inference in occlusion-aware generative models of images,
Jonathan Huang, Kevin Murphy.
ICLR Workshops, 2016

* It would be good to clarify that the paper is focusing on the image caption agreement task from Kuhnle and Copestake, as opposed to generic visual question answering.

* The claim that the paper works with natural language should be toned down and clarified. This is not natural language, firstly because the language in the dataset is synthetically generated and not “natural”. Secondly, the approach parses this “synthetic” language into structured tuples which makes it even less natural. Also, Page. 3. What does “partial descriptions” mean?

* Section 3: It would be good to explicitly draw out the graphical model for the proposed approach and clarify how it differs from prior work (Eslami, 2016).

* Sec. 3. 4 mentions that the “only image” encoder is used to obtain the representation for the image, but the “only image” encoder is expected to capture the “indescribable component” from the image, then how is the attribute information from the image captured in this framework? One cannot hope to do image caption association prediction without capturing the image attributes...

*, In general, the writing and presentation of the model seem highly fragmented, and it is not clear what the specifics of the overall model are. For instance, in the decoder, the paper mentions for the first time that there are variables “z”, but does not mention in the encoder how the variables “z” were obtained in the first place (Sec. 3.1). For instance, it is also not clear if the paper is modeling variable length sequences in a similar manner to Eslami, 2016 or not, and if this work also has a latent variable [z, z_pres] at every timestep which is used in a similar manner to Eqn. 2 in Eslami, 2016. Sec. 3.4 “GEN Image Encoder” has some typo, it is not clear what the conditioning is within q(z) term.

* Comparison to baselines:
  1. How well does this model do against a baseline discriminative image caption ranking approach, similar to [D]? This seems like an important baseline to report for the image caption ranking task.
  2. Another crucial baseline is to train the Attend, Infer, Repeat model on the ShapeWorld images, and then take the latent state inferred at every step by that model, and use those features instead of the features described in Sec. 3.4 “Gen Image Encoder” and repeat the rest of the proposed pipeline. Does the proposed approach still show gains over Attend Infer Repeat?
  3. The results shown in Fig. 7 are surprising -- in general, it does not seem like a regular VAE would do so poorly. Are the number of parameters in the proposed approach and the baseline VAE similar? Are the choices of decoder etc. similar? Did the model used for drawing Fig. 7 converge? Would be good to provide its training curve. Also, it would be good to evaluate the AIR model from Eslami, 2016 on the same simple shapes dataset and show unconditional samples. If the claim from the work is true, that model should be just as bad as a regular VAE and would clearly establish that using language is helping get better image samples.

* Page 2: In general the notion of separating the latent space into content and style, where we have labels for the “content” is an old idea that has appeared in the literature and should be cited accordingly. See [B] for an earlier treatment, and an extension by [A]. See also the Bivcca-private model of [C] which has “private” latent variables for vision similar to this work (this is relevant to Sec. 3.2.)

References:
[A]: Upchurch, Paul, Noah Snavely, and Kavita Bala. 2016. “From A to Z: Supervised Transfer of Style and Content Using Deep Neural Network Generators.” arXiv [cs.CV]. arXiv. http://arxiv.org/abs/1603.02003.

[B]: Kingma, Diederik P., Danilo J. Rezende, Shakir Mohamed, and Max Welling. 2014. “Semi-Supervised Learning with Deep Generative Models.” arXiv [cs.LG]. arXiv. http://arxiv.org/abs/1406.5298.

[C]: Wang, Weiran, Xinchen Yan, Honglak Lee, and Karen Livescu. 2016. “Deep Variational Canonical Correlation Analysis.” arXiv [cs.LG]. arXiv. http://arxiv.org/abs/1610.03454.

[D]: Kiros, Ryan, Ruslan Salakhutdinov, and Richard S. Zemel. 2014. “Unifying Visual-Semantic Embeddings with Multimodal Neural Language Models.” arXiv [cs.LG]. arXiv. http://arxiv.org/abs/1411.2539.

---

### Official Review · AnonReviewer1 · 2017-11-27
**GENERATIVE ENTITY NETWORKS: DISENTANGLING ENTITIES AND ATTRIBUTES IN VISUAL SCENES USING PARTIAL NATURAL LANGUAGE DESCRIPTIONS**

**Rating:** 5
**Confidence:** 5

**Review:**

Summary: The authors observe that the current image generation models generate realistic images however as the dimensions of the latent vector is fully entangled, small changes to a single neuron can effect every output pixel in arbitrary ways. In this work, they explore the effect of using partial natural language scene descriptions for the task of disentangling the latent entities visible in the image.  The proposed Generative Entity Networks jointly generates the natural language descriptions and images from scratch. The core model is Variational Autoencoders (VAE) with an integrated visual attention mechanism that also generates the associated text. The experiments are conducted on the Shapeworld dataset.

Strengths:
Simultaneous text and image generation is an interesting research topic that is relevant for the community.
The paper is well written, the model is formulated with no errors (although it could use some more detail) and supported by illustrations (although there are some issues with the illustrations detailed below).
The model is evaluated on tasks that it was not trained on which indicate that this model learns generalizable latent representations.

Weaknesses:
The paper gives the impression to be rushed, i.e. there are citations missing (page 3 and 6), the encoder model illustration is not as clear as it could be. Especially the white boxes have no labels, the experiments are conducted only on one small-scale proof of concept dataset, several relevant references are missing, e.g. GAN, DCGAN, GAWWN, StackGAN. Visual Question answering is mentioned several times in the paper, however no evaluations are done in this task.

Figure 2 is complex and confusing due to the lack of proper explanation in the text. The reader has to find out the connections between the textual description of the model and the figure themselves due to no reference to particular aspects of the figure at all. In addition the notation of the modules in the figure is almost completely disjoint so that it is initially unclear which terms are used interchangeably.

Details of the “white components” in Figure 2 are not mentioned at all. E.g., what is the purpose of the fully connected layers, why do the CNNs split and what is the difference in the two blocks (i.e. what is the reason for the addition small CNN block in one of the two)

The optimization procedure is unclear. What is the exact loss for each step in the recurrence of the outputs (according to Figure 5)? Or is only the final image and description optimized. If so, how is the partial language description as a target handled since the description for a different entity in an image might be valid, but not the current target. (This is based on my understanding that each data point consists of one image with multiple entities and one description that only refers to one of the entities).

An analysis or explanation of the following would be desirable: How is the network trained on single descriptions able to generate multiple descriptions during evaluation. How does thresholding mentioned in Figure 5 work?

In the text, k suggests to be identical to the number of entities in the image. In Figure 5, k seems to be larger than the number of entities. How is k chosen? Is it fixed or dynamic?

Even though the title claims that the model disentangles the latent space on an entity-level, it is not mentioned in the paper. Intuitively from Figure 5, the network generates black images (i.e. all values close to zero) whenever the attention is on no entity and, hence, when attention is on an entity the latent space represents only this entity and the image is generated only showing that particular entity. However, confirmation of this intuition is needed since this is a central claim of the paper.

As the main idea and the proposed model is simple and intuitive, the evaluation is quite important for this paper to be convincing. Shapeworlds dataset seems to be an interesting proof-of-concept dataset however it suffers from the following weaknesses that prevent the experiments from being convincing especially as they are not supported with more realistic setups. First, the visual data is composed of primitive shapes and colors in a black background. Second, the sentences are simple and non-realistic. Third, it is not used widely in the literature, therefore no benchmarks exist on this data.

It is not easy to read the figures in the experimental section, no walkthrough of the results are provided. For instance in Figure 4a, the task is described as “showing the changes in the attribute latent variables” which gives the impression that, e.g. for the first row the interpolation would be between a purple triangle to a purple rectangle however in the middle the intermediate shapes also are painted with a different color. It is not clear why the color in the middle changes.

The evaluation criteria reported on Table 1 is not clear. How is the accuracy measured, e.g. with respect to the number of objects mentioned in the sentence, the accuracy of the attribute values, the deviation from the ground truth sentence (if so, what is the evaluation metric)? No example sentences are provided for a qualitative comparisons. In fact, it is not clear if the model generates full sentences or attribute phrases.

As a summary, this paper would benefit significantly with a more extensive overview of the existing relevant models, clarification on the model details mentioned above and a more through experimental evaluation with more datasets and clear explanation of the findings.

---

### Author Response · Authors · 2018-01-05
**Thanks!**

We would like to thank the reviewers for reading the paper so carefully and for their detailed reviews. In addition to polishing the writing, and filling out the related work section, the main weakness of the paper seems to be both our evaluation on only one dataset, as well as comparing to only the dataset baselines rather than more recent stronger baselines.  We plan to work on this, and resubmit to a later conference.

However, we did want to clarify a few confusions which were brought up in the reviews.

---

> ### Author Response · Authors · 2018-01-05
> **RE: Reviewer 1**
>
> (a)	"Visual Question answering is mentioned several times in the paper, however no evaluations are done in this task"
>
> Most of the mentions of visual question answering in the paper are meant to refer to the general task of answering questions about an image, and in this sense, the Shapeworld caption classification we evaluate on is a visual question answering task.
>
> (b)	"The optimization procedure is unclear. What is the exact loss for each step in the recurrence of the outputs (according to Figure 5)? Or is only the final image and description optimized. If so, how is the partial language description as a target handled since the description for a different entity in an image might be valid, but not the current target. (This is based on my understanding that each data point consists of one image with multiple entities and one description that only refers to one of the entities)."
>
> We discuss this in Section 3.3 where we say: "However as the model outputs language predictions for multiple objects, and yet only one object is described in language per scene, we maximize over assignments of predicted language to the true caption."
>
> (c)	"An analysis or explanation of the following would be desirable: How is the network trained on single descriptions able to generate multiple descriptions during evaluation."
>
> The network always generates multiple descriptions (one for each recurrent step), but as we just highlighted, a loss signal is only generated between the provided description, and the generated description which most closely matches it.
>
> (d)	"How does thresholding mentioned in Figure 5 work?"
> For every encoded entity, the model decoder outputs word probabilities. We report all words assigned probability greater than 0.5  by the model.
>
> (e)	"In the text, k suggests to be identical to the number of entities in the image. In Figure 5, k seems to be larger than the number of entities. How is k chosen? Is it fixed or dynamic?"
> k is chosen statically as an upper bound on the number of entities in the image.  The model can then avoid using entities, by not drawing anything to the image for a given entity, and by generating zeros for all natural language attributes.
>
> (f)	"Even though the title claims that the model disentangles the latent space on an entity-level, it is not mentioned in the paper. Intuitively from Figure 5, the network generates black images (i.e. all values close to zero) whenever the attention is on no entity and, hence, when attention is on an entity the latent space represents only this entity and the image is generated only showing that particular entity. However, confirmation of this intuition is needed since this is a central claim of the paper."
>
> Figure 4(b)(d) indicates that manipulation of latent variables associated with a particular entity results in visual changes in only one object (e.g. the location / rotation of the green rectangle in 4d). This indicates that the latent representation is disentangled on an entity-level.  Did you have a specific experiment in mind that you thought would more clearly show that the representation was disentangled?
>
> (g)	"no benchmarks exist on this data"
> There do exist a carefully chosen set of benchmarks for the VQA dataset which were adapted for this dataset, and these are the benchmarks that we compare to. But we agree that benchmarks for generative modeling don't exist for this dataset.
>
> (h)	"It is not easy to read the figures in the experimental section, no walkthrough of the results are provided. For instance in Figure 4a, the task is described as showing the changes in the attribute latent variables which gives the impression that, e.g. for the first row the interpolation would be between a purple triangle to a purple rectangle however in the middle the intermediate shapes also are painted with a different color. It is not clear why the color in the middle changes."
>
> We attempted to address this issue in the figure caption where we said:  "Note that we should not expect the division between the color and shape semantic attributes to align to the two latent dimensions since the GEN model leaves the encoding of the attribute dimensions completely entangled for a given entity."
>
>
> (i)	"The evaluation criteria reported on Table 1 is not clear. How is the accuracy measured, e.g. with respect to the number of objects mentioned in the sentence, the accuracy of the attribute values, the deviation from the ground truth sentence (if so, what is the evaluation metric)? No example sentences are provided for a qualitative comparisons. In fact, it is not clear if the model generates full sentences or attribute phrases."
>
> We should have made this more clear in the paper.  Each natural language description in the dataset is labeled as either True or False, and the task is to predict this label.  So the accuracy numbers simply indicate whether or not a given description is correctly predicted to be True or False.

---

> ### Author Response · Authors · 2018-01-05
> **RE: Reviewer 2**
>
> (a)	"A major limitation of the model seems to be that one needs access to both images and attribute vectors at inference time to compute representations which is a highly restrictive assumption (since inference networks are discriminative). The paper should explain how/if one can compute representations given just the image, for instance, say by not using amortized inference. The paper does propose to use an image-only encoder but that is intended in general as a modeling choice to explain statistics which are not captured by the attributes (in this case location and orientation as explained in the Introduction of the paper)."
>
> There is a misunderstanding here and we should have clarified this in the paper. The model only needs access to an image as input in order to perform inference. In fact this is how we obtain the representations used in the auxilliary ShapeWorld tasks. We encode images without paired language by ignoring the language inputs in equation (5), instead using \hat{a} = a^{I}.
>
>
> (b)	"* Sec. 3. 4 mentions that the only image encoder is used to obtain the representation for the image, but the only image encoder is expected to capture the indescribable component from the image, then how is the attribute information from the image captured in this framework? One cannot hope to do image caption association prediction without capturing the image attributes..."
>
> As we discussed in section 3.1, the image encoder generates both the attribute information as well as the "indescribable component".  When the encoder is also provided the language, then the multi-modal aggregator is used to coherently combine the attribute predictions generated from the language with the predictions from the image encoder.
>
> (c)	"in the decoder, the paper mentions for the first time that there are variables z, but does not mention in the encoder how the variables z were obtained in the first place (Sec. 3.1).”
>
> We appreciate that this is not sufficiently explained in the paper and will clarify this in future work.  We did mention in Sec 3.1 the process for obtaining the visual latent variables z^V_k: "Finally each visual object representation v_k is passed through an MLP to obtain the parameters of the approximate posterior distribution over that object’s visual latent variables." However we did not state the equivalent process for obtaining the attribute latent variables z^A_k (which is achieved by applying a MLP to a^hat_k).
>
> (d)	“Sec. 3.4 GEN Image Encoder has some typo, it is not clear what the conditioning is within q(z) term.”
> Yes this is a typo, the notation should read q*(z | I), where q* is the encoder applied to only to input images with the modification described earlier.

---

### Decision · Program_Chairs · 2018-01-29
**ICLR 2018 Conference Acceptance Decision**

**Decision:**

Reject

**Comment:**

This paper presents a novel model for generating images and natural language descriptions simultaneously. The aim is to distangle representations learned for image generation by connecting them to the paired text. The reviews praise the problem setup and the mathematical formulation. However they point out significant issues with the clarity of the presentation in particular the diagrams, citations, and optimization procedure in general. They also point out issues with the experimental setup in terms of datasets used and lack of natural images for the tasks in question.  Reviews are impressively thorough and should be of use for a future submission.